# Quantitative Trait Loci and Candidate Genes That Control Seed Sugars Contents in the Soybean ‘Forrest’ by ‘Williams 82’ Recombinant Inbred Line Population

**DOI:** 10.3390/plants12193498

**Published:** 2023-10-08

**Authors:** Dounya Knizia, Nacer Bellaloui, Jiazheng Yuan, Naoufal Lakhssasi, Erdem Anil, Tri Vuong, Mohamed Embaby, Henry T. Nguyen, Alemu Mengistu, Khalid Meksem, My Abdelmajid Kassem

**Affiliations:** 1School of Agricultural Sciences, Southern Illinois University, Carbondale, IL 62901, USA; dounya.knizia@siu.edu (D.K.); naoufal.lakhssassi@siu.edu (N.L.); erdem.anil@siu.edu (E.A.); mohamed.embaby@siu.edu (M.E.); meksem@siu.edu (K.M.); 2USDA, Agriculture Research Service, Crop Genetics Research Unit, 141 Experiment Station Road, Stoneville, MS 38776, USA; nacer.bellaloui@usda.gov; 3Plant Genomics and Biotechnology Lab, Department of Biological and Forensic Sciences, Fayetteville State University, Fayetteville, NC 28301, USA; jyuan@uncfsu.edu; 4Division of Plant Science and Technology, University of Missouri, Columbia, MO 65211, USA; vuongt@missouri.edu (T.V.); nguyenhenry@missouri.edu (H.T.N.); 5USDA, Agriculture Research Service, Crop Genetics Research Unit, 605 Airways Blvd, Jackson, TN 38301, USA; alemu.mengistu@usda.gov

**Keywords:** soybean, RIL, Forrest, Williams 82, linkage map, RFOs, sucrose, raffinose, stachyose, SNPs

## Abstract

Soybean seed sugars are among the most abundant beneficial compounds for human and animal consumption in soybean seeds. Higher seed sugars such as sucrose are desirable as they contribute to taste and flavor in soy-based food. Therefore, the objectives of this study were to use the ‘Forrest’ by ‘Williams 82’ (F × W82) recombinant inbred line (RIL) soybean population (*n* = 309) to identify quantitative trait loci (QTLs) and candidate genes that control seed sugar (sucrose, stachyose, and raffinose) contents in two environments (North Carolina and Illinois) over two years (2018 and 2020). A total of 26 QTLs that control seed sugar contents were identified and mapped on 16 soybean chromosomes (chrs.). Interestingly, five QTL regions were identified in both locations, Illinois and North Carolina, in this study on chrs. 2, 5, 13, 17, and 20. Amongst 57 candidate genes identified in this study, 16 were located within 10 Megabase (MB) of the identified QTLs. Amongst them, a cluster of four genes involved in the sugars’ pathway was collocated within 6 MB of two QTLs that were detected in this study on chr. 17. Further functional validation of the identified genes could be beneficial in breeding programs to produce soybean lines with high beneficial sucrose and low raffinose family oligosaccharides.

## 1. Introduction

Sugars, including sucrose, stachyose, glucose, raffinose, galactose, fructose, rhamnose, and starch, play a major role in seed and fruit development and seed desiccation tolerance (DT) [1,2,3,4]. Sucrose and raffinosaccharides (raffinose and stachyose), also called raffinose family oligosaccharides (RFOs), make up 5–7%, 1%, and 3–4% of total carbohydrates, respectively, of soybean seed dry weights [5]. RFOs are synthesized from sucrose through a series of additions of galactinol units and are involved in DT, freezing, stress tolerance, and seed longevity [6,7,8,9]. Galactinol synthase (GolS) is the key enzyme in the RFO biosynthetic pathway converting galactinol and myo-inositol as the main precursors to form RFOs. Galactinol synthase (GolS) converts myo-inositol and UDP-galactose into galactinol, while sucrose and galactinol are converted into raffinose by raffinose synthase [9,10]. In addition to being involved in stress tolerance, RFOs are reported to play a role in several signal transduction pathways [11], exports of specific mRNAs [12], and trafficking of certain vesicle membranes [13].

Like most seed components, seed sugars [4] are influenced by many factors, including abiotic and biotic stresses, and environmental factors, such as temperature, soil moisture, freezing, seed maturity, and growth conditions [1,14,15,16,17,18,19]. It was shown that stachyose contents increased drastically in drying seeds but not in seeds kept at high humidity levels, which reveals the critical role of stachyose in DT [1]. The effect of water deficit (WD) on enzymes involved in sugar biosynthetic pathways in soybean nodules was investigated. Sucrose synthase activity declined drastically with increased WD while sucrose content increased [14]. Other studies showed that WD impacts negatively on sucrose biosynthesis and translocation from sources to sinks more than other sugars’ (raffinose and stachyose) biosynthesis [16,19]. Investigating ‘Clark’ and ‘Harosoy’ near-isogenic lines (NILs) revealed that Clark’s sugar contents decreased with increased days of maturity for both cultivars while both positive and negative effects were observed concerning the effects of temperature in two different years (2004 and 2005) [15]. In 2004, seed sugar contents increased with temperature increase, while the contents decreased with increased temperatures in 2005 [15]. The effect of WD on several seed components, including sugars, was investigated in several susceptible and resistant Phomopsis soybean cultivars. Sugar (sucrose, raffinose, and stachyose) contents were higher in seeds of resistant maturity group III cultivars than their susceptible counterparts [16]. A recent study investigated the effect of soil moisture on seed sugars (sucrose, raffinose, stachyose) and starch contents among other compounds in two soybean cultivars in maturity group V (Asgrow, AG6332, and Progeny 5333RY) and showed that sucrose, stachyose, and raffinose contents, in addition to the mineral nutrient (N, P, K, and Ca) contents, decreased with increased soil moisture in both cultivars [17].

During recent decades, more than 53 QTLs that control seed sucrose and RFOs, other sugars (glucose, galactose, fructose, fucose, rhamnose), and starch contents have been reported using different biparental and natural populations and mapping methods including single marker analysis, interval mapping (IM), composite interval mapping (CIM), and genome-wide association studies (GWASs) [18,20]. However, to our knowledge, only a few of these studies identified candidate genes within these QTL regions, as summarized in [18]. There is *Glyma.01g127600*, which encodes for a protein phosphatase on chr. 1; *Glyma.03g019300*, which encodes for a MADS-box protein; *Glyma.03g064700*, which encodes for a phosphatidylinositol monophosphate-5-kinase on chr. 3; and *Glyma.06g194200*, which encodes for a gibberellin-regulated protein on chr. 6 [18,21].

To improve seed quality, several attempts to manipulate seed sugars, phytic acid, and the contents of other beneficial compounds have been made in recent years [22,23,24]. Monogastric animals (such as poultry and pigs) and humans do not produce α-galactosidase and cannot digest RFOs, which reduces gastrointestinal performance, flatulence, and diarrhea. Therefore, reducing raffinose and stachyose and increasing sucrose in soybean seed contents are desirable and the main goals in breeding programs [22,23,24,25,26,27]. The objective of this study was to genetically map QTLs for seed sucrose, raffinose, and stachyose contents using the ‘Forrest’ by ‘Williams 82’ RIL population, in addition to identifying candidate genes involved in soybean seed sugar biosynthesis.

## 2. Materials and Methods

### 2.1. Plant Materials

The ‘Forrest’ × ‘Williams 82’ RIL population (F × W82, *n* = 309) was previously studied and described in detail in our previous research [28,29]. The parents and RILs were evaluated in two locations: Spring Lake, NC (35.17° N, 78.97° W, 2018) and Carbondale, IL (37° N, 89° W, 2020). Briefly, seed parents and RIL seeds were grown in a randomized block design with 25 cm row spaces and three replicates. More details about growth conditions, crop management, and seed harvesting were described earlier [28,29].

### 2.2. Seed Sugar Quantification

RILs, parents (Forrest and Williams 82), and soybean germplasm seeds were harvested at maturity, and sugar (sucrose, raffinose, and stachyose) contents (%) were quantified using near-infrared reflectance (NIR) with an AD 7200 array feed analyzer (Perten, Springfield, IL, USA) as described earlier [15,30].

### 2.3. DNA Isolation, SNP Genotyping, and Genetic Map Construction

Parents’ and RILs’ genomic DNA was extracted using the cetyltrimethylammonium bromide (CTAB) method as previously described [31]. A NanoDrop spectrophotometer (NanoDrop Technologies Inc., Centreville, DE, USA) was used to quantify DNA samples (50 ng/µL), and genotyping was performed using the Illumina Infinium SoySNP6K BeadChips (Illumina, Inc., San Diego, CA, USA) as described earlier [15] at the Soybean Genomics and Improvement Laboratory (USDA-ARS, Beltsville, MD, USA). The F × W82 genetic linkage map was constructed using JoinMap 4.0 [28,32] as previously described to detect QTLs for seed isoflavones [28] and seed tocopherol contents [29].

### 2.4. Sugar QTL Detection

WinQTL Cartographer [33] interval mapping (IM) and composite interval mapping (CIM) methods were used to identify QTLs that control seed sugar (sucrose, stachyose, and raffinose) contents in this RIL population. The following parameters were used: Model 6, 1 cM step size, 10 cM window size, 5 control markers, and 1000 permutations. Furthermore, chromosomes were drawn using MapChart 2.2 [34].

### 2.5. Sugars Biosynthesis Candidate Genes’ Identification

The Glyma numbers of the sucrose and RFO biosynthesis genes were obtained via reverse BLAST of the genes underlying the RFO pathway in *Arabidopsis* using the available data in SoyBase. The sequences of the *Arabidopsis* genes were obtained from the Phytozome database (https://phytozome-next.jgi.doe.gov; accessed on 15 August 2023). These sequences were used for Blast in SoyBase. The obtained genes that control the RFO pathway were mapped to the identified sugars’ QTLs.

### 2.6. Expression Analysis

The expression analysis of the identified candidate genes was performed using the publicly available data from SoyBase [20] to produce the expression profiles of these candidate genes in the soybean reference genome Williams 82 in the Glyma1.0 Gene Models version.

### 2.7. Comparison of the Williams 82 and Forrest Sequences

Sequences of Forrest and Williams 82 cv. were obtained from the variant calling and haplotyping analysis, which was performed using 108 soybean germplasm sequenced lines as described previously [35].

## 3. Results

### 3.1. Sugar Frequency Distribution

The frequency distributions among sucrose, raffinose, and stachyose contents were quite different in the F × W82 RIL population based on the Shapiro–Wilk method for the normality test. Raffinose (2018), stachyose (2018), and sucrose (2020) were normally distributed. Only positive or negative skewness were identified in the RIL population, and all kurtosis values of these variables were positive (Table 1; Figure 1). In terms of coefficient of variation (CV), the value of sucrose 2018 showed the highest percentage of variation (62.86%) compared to other assessed traits, and the rest of the CVs appeared to be less varied within these two years. The histogram of sucrose 2018 was extremely skewed, and the other traits evaluated were normally distributed.

The broad-sense heritability (*h*^2^*_b_*) of seed sugar weight for sucrose, raffinose, and stachyose contents across two different environments appeared quite different. Stachyose had the highest heritability (92%), and the *h*^2^*_b_* for sucrose was 36.8% (Table 2). However, no negative *h*^2^*_b_* values for sugar contents were observed. The RIL–year interactions still played a significant role in the molecular formation among the three sugar contents in soybean seeds. The Sum Sq and Mean Sq to determine σ_G_^2^ and σ_GE_^2^ for each trait (Table 2) using the type I sum of squares (ANOVA (model)) function in the R program were implemented.

### 3.2. Sugars Contents’ QTLs

IM and CIM were used to identify QTLs for seed sugar contents in this F × W82 RIL population; however, only QTLs identified by CIM are presented here. The QTLs identified with the IM method are reported in Appendix A. A total of 26 QTLs that control seed sugar contents were identified in both NC-2018 (19 QTLs) and IL-2020 (7 QTLs) via CIM (Table 3 and Table 4; Appendix A).

In Spring Lake, NC in 2018 (NC-2018), 12 QTLs that control seed sucrose content (qSUC-1–qSUC-12) were identified and mapped on Chrs. 1, 2, 3, 4, 5, 6, 9, 10, 13, 17, 18, and 19; 4 QTLs that control seed stachyose content (qSTA-1–qSTA-4) were identified and mapped on Chrs. 13 and 19; and 3 QTLs that control seed raffinose content (qRAF-1–qRAF-3) were identified and mapped on Chr. 9 and 12 (Table 3 and Table 5; Appendix A). Likewise, in Carbondale, IL in 2020 (IL-2020), 3 QTLs that control seed sucrose content (qSUC-1–qSUC-3) were identified and mapped on Chrs. 2, 5, and 8; and 4 QTLs that control seed stachyose content (qSTA-1–qSTA-4) were identified and mapped on Chrs. 13, 16, 17, and 20 (Table 4 and Table 6; Appendix A). No QTL that controls seed raffinose content was identified in this location. 

No QTL for seed sugar contents was identified in other studies within the QTL regions on chr. 4 (qSUC-4-NC-2018, 6.5–16.5 cM), chr. 10 (qSUC-8-NC-2018, 214.1–216.1 cM), or chr. 18 (qSUC-11-NC-2018, 20.1–17.5 cM), which indicates they are novel QTL regions.

### 3.3. In Silico Sucrose, Raffinose, and Stachyose Biosynthetic Pathway Genes in Soybean

In the literature, the sugar (sucrose, raffinose, and stachyose) biosynthetic pathway was studied in many plants, including the plant model *Arabidopsis thaliana* [36,37] and the leguminous model *Medicago sativa* L. [38]. A reverse BLAST of the genes identified in *Arabidopsis thaliana* was conducted using SoyBase [20] to reconstruct the sugar (sucrose, raffinose, and stachyose) biosynthetic pathway in soybean (Figure 2).

A total of fifty-seven candidate genes were identified to underly the sugar (sucrose, raffinose, and stachyose) biosynthetic pathway (Figure 2). In this pathway, twelve candidate genes were identified for invertase: *Glyma.05G185500*, *Glyma.20G177200*, *Glyma.08G043800*, *Glyma.10G214700*, *Glyma.08G143500*, *Glyma.05G236600*, *Glyma.17G037400*, *Glyma.10G145600*, *Glyma.20G095200*, *Glyma.07G236000*, *Glyma.02G016700*, and *Glyma.10G017300*. Twelve candidate genes were identified for sucrose synthase: *Glyma.02G240400*, *Glyma.03G216300*, *Glyma.09G073600*, *Glyma.09G167000*, *Glyma.13G114000*, *Glyma.14G209900*, *Glyma.15G151000*, *Glyma.16G217200*, *Glyma.17G045800*, *Glyma.19G212800*, *Glyma.11G212700*, and *Glyma.15G182600*. Twelve candidate genes were identified for UDP-D-Glucose-4-Epimerase: *Glyma.08G023100*, *Glyma.01G225800*, *Glyma.05G204700*, *Glyma.05G217100*, *Glyma.07G237700*, *Glyma.07G271200*, *Glyma.08G011800*, *Glyma.11G017100*, *Glyma.12G162600*, *Glyma.17G035800*, *Glyma.18G145700*, and *Glyma.18G211700.* For galactinol synthase, six candidate genes were identified: *Glyma.03G222000*, *Glyma.03G229800*, *Glyma.10G145300*, *Glyma.19G219100*, *Glyma.19G227800*, and *Glyma.20G094500.* Fourteen candidate genes were identified for raffinose synthase: *Glyma.03G137900*, *Glyma.04G145800*, *Glyma.19G140700*, *Glyma.04G190000*, *Glyma.02G303300*, *Glyma.05G003900*, *Glyma.06G175500*, *Glyma.09G016600*, *Glyma.13G160100*, *Glyma.14G010500*, *Glyma.17G111400*, *Glyma.19G004400*, *Glyma.05G040300*, and *Glyma.06G179200*. For stachyose synthase, only one candidate gene was identified: *Glyma.19G217700* (Figure 2).

### 3.4. Association between the Identified Sugar (Sucrose, Raffinose, and Stachyose) Biosynthetic Pathway Candidate Genes and Reported QTLs

The identified genes for sugar (sucrose, raffinose, and stachyose) biosynthesis in soybean were mapped to the identified QTLs. Amongst fifty-seven candidate genes, sixteen were located less than 10 MB from the identified QTLs on chrs. 2, 5, 6, 8, 9, 10, 17, and 19 (Table 3, Table 4, Table 5 and Table 6).

The sucrose synthase candidate gene *Glyma.09G073600* and the raffinose synthase candidate gene *Glyma.09G016600* are positioned close to *qSUC-7-IL-2018*, *qRAF-1-IL-2018*, and *qRAF-2-IL-2018* on Chr.9 (Table 3, Table 4, Table 5 and Table 6). The invertase candidate gene *Glyma.02G016700* is located 3.6 and 0.2 MB away from *qSUC-1-IL-2018* and *qSUC-1-NC-2020,* respectively, on Chr. 2 (Table 3, Table 4, Table 5 and Table 6). The raffinose synthase candidate genes *Glyma.05G003900* and *Glyma.05G040300* are located close to *qSUC-5-IL-2018* and *qSUC-2-NC-2020* on Chr. 5 (Table 3, Table 4, Table 5 and Table 6). On chr. 6, the raffinose synthase candidate gene *Glyma.06G175500* is located close to *qSUC-6-IL-2018* (Table 3, Table 4, Table 5 and Table 6). The invertase candidate genes *Glyma.08G043800* and *Glyma.08G143500*, and the UDP-D-Glucose-4-Epimerase candidate genes *Glyma.08G011800* and *Glyma.08G023100* on chr. 8 are located close to *qSUC-3-NC-2020* (Table 3, Table 4, Table 5 and Table 6, Appendix A). On chr. 10, the invertase candidate gene *Glyma.10G017300* is located close to *qSUC-8-IL-2018* (Table 3, Table 4, Table 5 and Table 6). On Chr. 17, a cluster of four genes involved in the sugar pathway is collocated within 6 MB of two QTLs (qSUC-10-NC-2018 and qSTA-3-IL-2020) that were identified in this study. These genes are *Glyma.17G037400* encoding for an invertase, *Glyma.17G045800* encoding for sucrose synthase, *Glyma.17G111400* encoding for raffinose synthase, and *Glyma.17G035800* encoding for UDP-D-glucose-4-epimerase (Table 3, Table 4, Table 5 and Table 6, Appendix A). The raffinose synthase candidate gene *Glyma.19G004400* is positioned close to *qSTA-3-IL-2018* and *qSTA-4-IL-2018* (Table 3, Table 4, Table 5 and Table 6), as well as *qRAF-8-IL-2018* and *qRAF-9-IL-2018* identified using the IM method (Table 3 and Table 4).

### 3.5. Association between the Identified Candidate Genes and the Previously Reported QTLs

Several studies have identified and mapped QTLs underlying the seed sugar content using different populations and mapping methods [39,40,41,42], as summarized in [18].

The identified genes have been mapped to the previously reported QTL regions associated with the seed sugar content using data from SoyBase [18,20,43]. In this study, 6 candidate genes were located within the identified seed sugar QTLs and 18 were located <9 MB away from these regions (Table 7). Among them is the invertase candidate gene *Glyma.08G143500*, which is located within the seed sucrose 1-2 QTL on Chr. 8 [20,39]. Also, the galactinol-sucrose galactosyl-transferase 6-related candidate gene *Glyma.13G160100* is situated within the seed sucrose 1-5 QTL [20,39] (Table 7). Likewise, the raffinose synthase candidate gene *Glyma.19G140700* is collocated within the seed sucrose 1-8 QTL [20,39], less than <0.5 MB away from seed sucrose 2-11 and seed sucrose 2-10 [20,41], and 1.9 MB from seed oligosaccharide 2-7 [20,40].

The sucrose synthase candidate gene *Glyma.02G240400* was located close (<1.5 MB) to two QTLs controlling seed sugar contents, the seed sucrose 2-2 and seed oligosaccharide 1-1 [20,41]. Moreover, the raffinose synthase candidate gene *Glyma.05G003900* is located less than <4 MB away from the seed sucrose 1-1 [20,39]. The raffinose synthase candidate gene *Glyma.19G004400* is located less than 9 MB away from four QTLs controlling the sugar contents, namely seed sucrose 2-3, seed oligosaccharide 1-2, seed sucrose 2-6, and seed oligosaccharide 1-5 [20,41] (Table 7). On chr. 8, the seed sucrose 1-3 and seed sucrose 1-13 are located close to the invertase candidate genes *Glyma.08G043800*, and *Glyma.08G143500*, as well as UDP-D-glucose-4-epimerase candidate genes *Glyma.08G011800* and *Glyma.08G023100* [20,39] (Table 7). The sucrose synthase candidate gene *Glyma.09G073600* and the raffinose candidate gene *Glyma.09G016600* are positioned less than <2 MB away from the seed sucrose 4-2 [20,44] (Table 7). Interestingly, the sucrose synthase candidate genes *Glyma.15G182600* and *Glyma.15G151000* are located less than <1.25 MB from the seed sucrose 3-3 and seed oligosaccharide 2-3 [20,40].

### 3.6. Organ-Specific Expression of the Identified Candidate Genes

The expression pattern of the identified candidate genes was investigated in Williams 82 cv. using the RNA-seq data available in SoyBase [20]. The dataset includes several plant tissues, including leaves, nodules, roots, pods, and seeds (Figure 3A,B and Appendix A). Four of the fifty-seven identified candidate genes have no available RNA-seq data, including the sucrose synthase candidate genes *Glyma.03G216300*, *Glyma.17G045800*, and *Glyma.19G212800*, as well as the UDP-D-glucose-4-epimerase candidate gene *Glyma.18G211700* (Appendix A). The raffinose synthase candidate gene *Glyma.04G145800* was not expressed in any of the analyzed tissues, whilst the rest of the identified genes showed different expression patterns.

The sucrose synthase candidate genes *Glyma.09G073600* and *Glyma.13G114000* presented a high expression profile in all the analyzed tissues except for the young leaves, while the raffinose synthase candidate gene *Glyma.17G111400* was abundantly expressed in all the analyzed tissues except for the seeds and nodules. Interestingly, the sucrose synthase candidate gene *Glyma.15G182600* was highly expressed in all the tissues excluding the young leaves and the nodules. The raffinose synthase candidate gene *Glyma.03G137900* was abundantly expressed in flowers, nodules, and roots. The raffinose synthase candidate gene *Glyma.14G010500* and the invertase candidate gene *Glyma.05G236600* were highly expressed in the flowers and pods. Also, the UDP-D-glucose-4-epimerase candidate gene *Glyma.05G204700* was abundantly expressed in the flowers and seeds. While the invertase candidate gene *Glyma.20G177200* was highly expressed in nodules and roots, the raffinose synthase candidate gene *Glyma.06G179200* was found to be highly expressed in seeds (Figure 3A and Appendix A).

Seventeen of the identified candidate genes were situated less than 10 MB away from the identified QTL regions. *Glyma.09G073600* was highly expressed in seeds in Williams 82 cv., followed by *Glyma.17G111400*, *Glyma.17G035800*, and *Glyma.08G043800* with a moderated expression profile. The remaining genes had lower expression patterns, excluding the *Glyma.02G016700*, *Glyma.06G175500*, *Glyma.09G016600*, *Glyma.10G017300*, and *Glyma.19G004400* genes, which were not expressed in seeds in Williams 82 cv.

## 4. Comparison of the Williams 82 and Forrest Sequences

The sequences of the candidate genes located less than 10 MB from the identified QTLs were compared. The results showed that six of them had SNPs and InDels between the Forrest and Williams 82 sequences: *Glyma.09G073600*, *Glyma.08G143500*, *Glyma.05G003900*, *Glyma.17G035800*, *Glyma.17G111400*, and *Glyma.09G016600* (Appendix A, Figure 4).

The sucrose synthase *Glyma.09G073600* had in total 28 SNPs and 7 InDels; three of these SNPs were located upstream of the 5′UTR, two are downstream of the 3′UTR, and seven were located in the exons (Appendix A, Figure 4). For the invertase candidate gene *Glyma.08G143500,* there were 20 SNPs and 5 InDels. One of these SNPs was located in exon 7, causing a missense mutation, and two SNPs were located upstream of the 5′UTR (Appendix A, Figure 4). The raffinose synthase candidate gene *Glyma.05G003900* had nine SNPs and one InDel; four of those SNPs were in the exons, from which two SNPs resulted in missense mutations (Appendix A, Figure 4). Likewise, the raffinose synthase candidate gene *Glyma.09G016600* possessed 12 SNPs and 2 InDels. Amongst these SNPs, there were two located in exons, which resulted in missense mutations, in addition to the two InDels located in the exons (Appendix A, Figure 4). For the raffinose candidate gene *Glyma.17G111400*, eight SNPs were found, of which one was located upstream of the 5′ UTR, another one was downstream of the 3′UTR, and the last six were in exons causing silent mutations (Appendix A, Figure 4). Finally, the UDP-D-Glucose-4-Epimerase candidate gene *Glyma.17G035800* had two SNPs that were positioned in introns (Appendix A).

## 5. Discussion

Soybean seed sugars play a major role in seed and fruit development. Recently, soy products became very popular as a result of a growing demand for vegan diets [45]. The quality and taste of these products are determined by the soybean seed sugar content [39]. These sugars include sucrose, raffinose, and stachyose which make up 5–7%, 1%, and 3–4% of total carbohydrates, respectively [5]. However, the raffinose and stachyose fermentation by human and monogastric animal intestine microbes leads to a reduced gastrointestinal performance, flatulence, and diarrhea. Thus, reducing raffinose and stachyose and increasing sucrose in soybean seed content are desirable [22,27].

Given the importance of the soybean seed sucrose content for the quality of soybean-based products for food and feed, breeding programs are focused on developing soybean seeds with a high sucrose content and low RFO content [43,46]. Thus, soybean varieties with a high sucrose content are valuable for soybean food and feed products [47].

The identification of QTLs associated with sugar components using different types of molecular markers is one of the breeding-process approaches that researchers use to breed for a high-sucrose soybean. In soybean and other crops, it is well established that seed sugar contents are complex polygenic traits, and many studies including this study have mapped QTLs for sugar contents using various mapping populations including biparental populations where parents may not necessarily have contrasting amounts of sugar contents, such as in the “MD96-5722” by “Spencer” RIL population [30].

In the current study, all seed sugar (sucrose, raffinose, and stachyose) phenotypic data, except one (sucrose, 2018), exhibited normal distributions in all environments studied (years and locations), showing that these traits are polygenic and complex, as shown previously [21,39,40,41,44,47,48,49,50,51,52,53].

The SNP-based genetic linkage map facilitated the identification of several QTLs including QTLs for seed isoflavone contents [28], seed tocopherol contents [29], and seed sugar (sucrose, stachyose, and raffinose) contents, as reported in the current study.

The heritability (H^2^) of sucrose, stachyose, and raffinose was estimated to be 37.5%, 73.9%, and 92%, respectively. There is no doubt that the environment and the interactions of genotype and environment play a major role in the heritability of traits [28,29,43,54,55]. A trait biosynthesis that involves several genes is expected to have a lower heritability than a trait biosynthesis that involves fewer genes. Figure 2 shows the number of potential genes that are involved in sucrose biosynthesis versus those involved in raffinose and stachyose; it seems like there is a correlation between the heritability values and the number of genes involved in the biosynthesis pathway.

Among the identified sugar QTLs, there are novel QTL regions (qSUC-4, qSUC-8, and qSUC-11). All the other QTLs have been located within or very close to the previously reported sugar QTLs [30,39,40,41,44], as summarized in [18]. Five other genomic regions on chrs. 2, 6, 12, 16, and 19 harboring sugar QTLs either from this study or from other studies are of particular interest. On chr. 2, qSUC-2-NC-2018 may correspond to *suc 1-1* identified previously [39]. This QTL region contains the *Glyma.02G016700* candidate gene that encodes for invertase.

Interestingly, several QTLs have been identified previously, including a QTL that controls seed raffinose content within the qSUC-1-NC-2018 region (chr. 1) [30], two QTLs (suc 2-2 and suc 3-2) that control seed sucrose content within the qSUC-2-NC-2018 region (chr. 2) [20,40,41], a QTL that controls seed sucrose content (suc-001) within the qSUC-3-NC-2018 region (chr. 3), [30], 2 QTLs that control seed sucrose (suc 1-1 and suc 4-1) content within the qSUC-5-NC-2018 region (chr. 5) [39,44], a QTL that controls seed raffinose content (raf003 and raf004) within the qSUC-6-NC-2018 and qSUC-7-NC-2018 regions (chrs. 6 and 9) [30], a QTL that controls seed sucrose (suc 1-5) content within the qSUC-9-NC-2018 region (chr. 13) [39], and a QTL that controls seed sucrose (suc 1-4) content within the qSUC-12-NC-2018 region (chr. 20) [39].

Likewise, several other QTLs have been identified previously: a QTL that controls seed sucrose (suc 2-2, 3-2) content within the qSUC-1-IL-2020 region (chr. 2) [40,41], a QTL that control seed sucrose (suc 1-1, 4-1) content within the qSUC-2-IL-2020 (chr. 5) [39,44] and qSUC-3-IL-2020 (chr. 8) regions, and a QTL that control seed sucrose (suc 1-2, 1-3, 1-13) content within the qSUC-3-IL-2020 region (chr. 8) [39]. Within the QTL regions that were found to control seed stachyose contents (qSTA-1-IL-2020, qSTA-2-IL-2020, and qSTA-4-IL-2020) reported in the current study on chrs. 13, 16, and 19, several QTLs that control seed sucrose (suc 1-4, 1-5, 3-5, 3-6) and seed raffinose (raff007) contents have been identified previously [39,40,41].

On chr. 6, qSUC-6-NC-2018 most likely corresponds to *suc 2-2* [41] and raffinose (*raf003*) QTL regions identified previously [30,39]. The QTL region contains the *Glyma.06G175500* candidate gene encoding for raffinose synthase. Interestingly, the genomic region on chr. 19 comprising a cluster of sucrose QTLs (suc 1-6 to 1-8, 2-3 to 2-11) [39,41] also contains two stachyose QTLs identified in this study (qSTA-3-NC-2018 and qSTA-4-NC-2018). The candidate gene *Glyma.19G004400*, which also encodes for raffinose synthase, was identified within this QTL region.

No candidate genes have been identified on chrs. 12 (qRAF-3-NC-2018), 16 (qSTA-2-NC-2018), or 20 (qSTA-4-NC-2018).

Remarkably, within the novel QTL regions reported here on chrs. 4, 10, and 18, seven candidate genes were identified, including the *Glyma.18G145700* encoding for UDP-D-glucose-4-epimerase on chr. 18 (Table 5 and Table 6, and Figure 2).

Interestingly, five QTL regions were detected in both locations, IL and NC. The first QTL region contains qSUC-5-NC-2018 and qSUC-2-IL-2020, which were detected in the same location on chr. 5. Additionally, qSUC-9-NC-2018, qSTA-1-NC-2018, and qSTA-2-NC-2018 were located only 1 MB away from qSTA-1-IL-2020 on chr.13. Moreover, qSUC-12-NC-2018 was 1.3 MB away from qSTA-4-IL-2020 on chr. 20. Furthermore, qSUC-10-NC-2018 and qSTA-3-IL-2020 were positioned 3.1 MB away from each other on chr. 17. Additionally, qSUC-2-NC-2018 and qSUC-1-IL-2020 were located ~4 MB away on chr. 2. The QTL regions that were not detected in both locations may be affected by environmental conditions.

In a previous study [54], 31,245 SNPs and 323 soybean germplasm accessions grown in three different environments were used to identify 72 QTLs associated with individual sugars and 14 associated with total sugar [54]. In addition, ten candidate genes that are within the 100 Kb flanking regions of the lead SNPs in six chromosomes were significantly associated with sugar content in soybean, eight of which are involved in the sugar metabolism in soybean [54]. Amongst these candidate genes, the raffinose synthase gene *Glyma.05G003900* was also reported in this study.

A recent study used an RIL population from a cross of ZD27 by HF25 to identify 16 QTLs controlling seed sucrose and soluble sugar contents in soybean [43]. Amongst these QTLs, qSU1701 [43] with an LOD = 7.61 and phenotypic variation explained (PVE) = 16.8% was identified on chr. 17 to be associated with the seed sucrose content. This QTL region is collocated with qSUC-10-NC-2018 identified in this study for the same trait with an LOD = 33.2 and an R^2^ = 20.5. On the same chr., qSS1701 [43] and qSS1702, identified to be associated with the seed soluble sugar content, are collocated with qSTA-3-IL-2020. These QTLs are positioned less than 8 MB away from a cluster of four genes involved in the sugars’ pathway, including *Glyma.17G037400* encoding for invertase, *Glyma.17G045800* encoding for sucrose synthase, *Glyma.17G111400* encoding for raffinose synthase (showing 7 SNP variations in exons) (Figure 4), and *Glyma.17G035800* encoding for UDP-D-glucose-4-epimerase. Our results confirm that this region on chr. 17 is a major QTL associated with seed sugar contents in soybean. In the same study [43], qSU2001 identified on chr. 20 with LOD = 3.38 and PVE = 5.6% was collocated with qSUC-12-NC-2018, and it was 0.3 MB away from qSTA-4-IL-2020. The invertase candidate gene Glyma.20G177200 is positioned within qSU2002 [43] identified on chr. 20 with LOD = 7.9 and PVE = 14.4%. These results confirm that this region on chr. 20 is involved in soybean seed sugar contents. On chr. 3, qSS0301 was previously identified [43] to be associated with soluble sugar contents in soybean with an LOD = 5.2 and PVE = 11.8. This QTL is located 1.4 MB away from qSUC-3-NC-2018.

The sucrose synthase gene *Glyma.09G073600* was highly expressed in seeds, followed by *Glyma.17G111400*, *Glyma.17G035800*, and *Glyma.08G043800* with moderated expression patterns in seeds. *Glyma.09G073600* and *Glyma.09G016600* are located close to qSUC-7-IL-2018, qRAF-1-IL-2018, and qRAF-2-IL-2018 on chr. 9. *Glyma.08G143500* is located close to qSUC-3-NC-2020, and *Glyma.05G003900* is positioned close to qSUC-5-IL-2018 and qSUC-2-NC-2020 on chr. 5. These genes could be useful in gene editing technology or breeding programs to develop soybean cultivars with reduced amounts of RFOs and high amounts of sucrose, which is beneficial for human consumption and animal feed.

Further studies are needed to characterize these genes, identify their enzymes and protein products, and understand their roles in the sugar biosynthetic pathway in soybean.

## 6. Conclusions

In summary, we have identified 26 QTLs associated with the seed sugar contents and 57 candidate genes involved in the sucrose, raffinose, and stachyose biosynthetic pathway. Amongst these candidate genes, 16 were located less than 10 MB away from the QTL regions identified in this study.

On chr. 17, a cluster of four genes controlling the sugar pathway is collocated within 6 MB of two QTLs (*qSUC-10-NC-2018* and *qSTA-3-IL-2020*) that were identified in this study. Moreover, the raffinose synthase candidate gene *Glyma.06G175500* is 9.7MB away from qSUC-6-NC-2018 QTL on chr. 6. The invertase candidate gene *Glyma.02G016700* is located 3.6 and 0.2 MB away from qSUC-1-NC-2018 (R^2^ = 47.9) and *qSUC-1-IL-2020* (R^2^ = 3.6), respectively, on chr. 2. Moreover, the sucrose synthase candidate gene *Glyma.09G073600* and the raffinose synthase candidate gene *Glyma.09G016600* were found close to qSUC-7-IL-2018, qRAF-1-IL-2018, qRAF-2-IL-2018, and qRAF-1-IL-2018 on chr. 9.

Five QTL regions were commonly identified in the two environments, NC and IL, on chrs. 2, 5, 13, 17 and 20 ((qSUC-5-NC-2018 and qSUC-2-IL-2020), (qSUC-9-NC-2018, qSTA-1-NC-2018, and qSTA-1-IL-2020), (qSUC-12-NC-2018 and qSTA-4-IL-2020), (qSUC-10-NC-2018 and qSTA-3-IL-2020), and (qSUC-2-NC-2018 and qSUC-1-IL-2020)).

Five genes (*Glyma.09G073600, Glyma.08G143500, Glyma.17G111400, Glyma.05G003900,* and *Glyma.09G016600*) have SNPs and InDels between the Forrest and Williams 82 sequences. These SNPs could potentially explain the difference in sugar content between Forrest and Williams 82 cultivars.

Further studies are required to functionally characterize these genes so we can understand and validate their roles in the sugar biosynthetic pathway in soybean before they are used in breeding programs to produce soybean lines with high beneficial sucrose and low RFOs.

## Figures and Tables

**Figure 1 plants-12-03498-f001:**
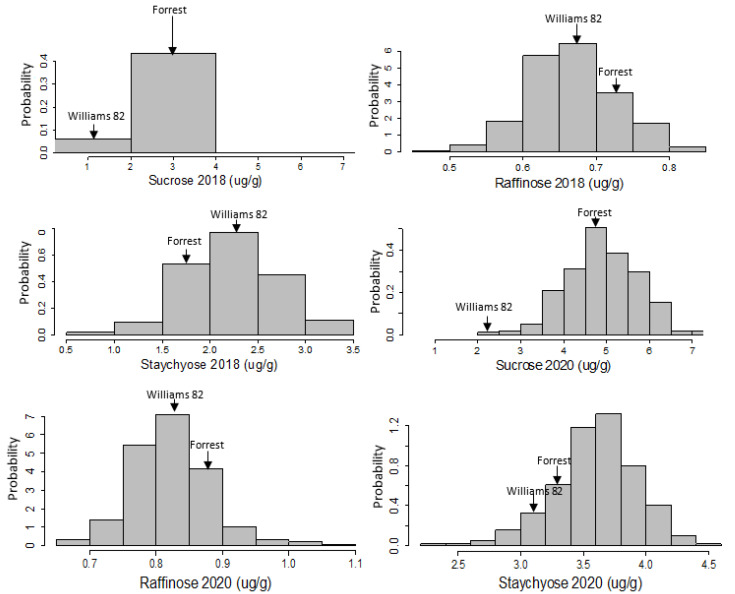
Frequency distribution of sugars (sucrose, raffinose, and stachyose) in the F × W82 RIL population grown in two environments over two years (Spring Lake, NC in 2018 and Carbondale, IL in 2020).

**Figure 2 plants-12-03498-f002:**
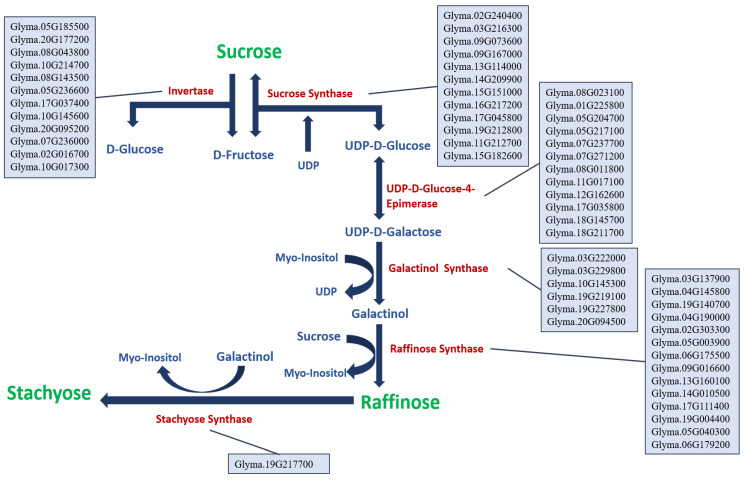
The sugar (sucrose, raffinose, and stachyose) biosynthetic pathway with the identified candidate genes in soybean. The genes are in Wm82.a2.v1 annotation.

**Figure 3 plants-12-03498-f003:**
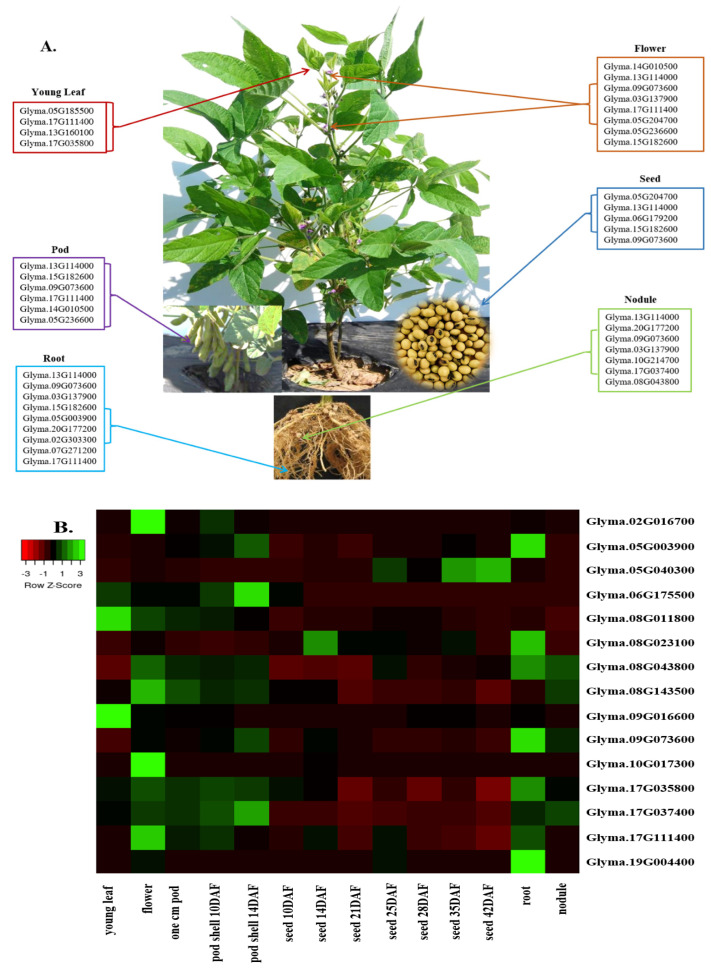
(**A**) Tissue-specific expression of the identified sugar candidate genes. (**B**) Expression HeatMap of the identified candidate genes located within 10 MB of the identified sugar QTL regions in Williams 82 (RPKM) were retrieved from publicly available RNA-seq data from the Soybase database [20]. RNA-seq data are not available in Soybase for the *Glyma.17G045800* candidate gene.

**Figure 4 plants-12-03498-f004:**
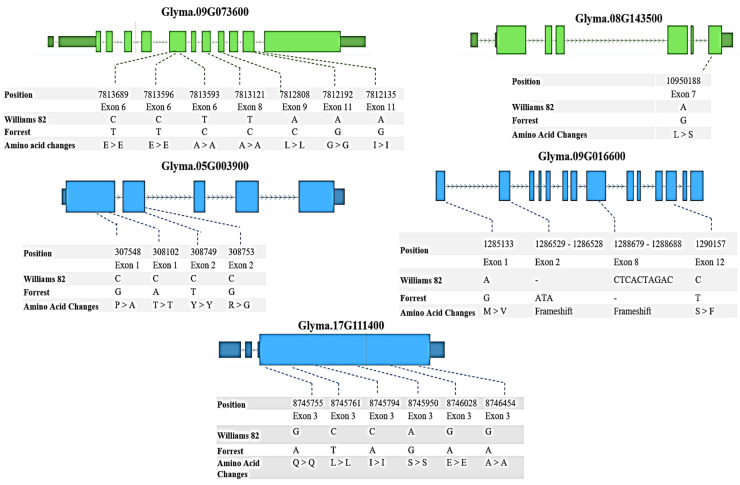
Positions of SNPs between Forrest and Williams 82 cultivars in *Glyma.09G073600*, *Glyma.08G143500*, *Glyma.05G003900*, *Glyma.17G111400*, and *Glyma.09G016600* coding sequences. In the gene model diagram, the light blue/light green boxes represent exons, blue/green bars represent introns, and dark blue/dark green boxes represent 3′UTR or 5′UTR. SNPs were positioned relative to the genomic position in the genome version W82.a2.

**Table 1 plants-12-03498-t001:** Seed sugar contents’ means, ranges, CVs, skewness, and kurtosis in the F × W82 RIL population evaluated in Spring Lake, NC (2018) and Carbondale, IL (2020). Mean and range values are expressed in µg/g of seed weight. ** *p* < 0.01, *** *p* < 0.001.

Year	Sugar	Mean	Range	CV (%)	SE	Skewness	Kurtosis	W Value (*p* < 0.05)
2018	Sucrose	2.58	22.7	62.86	0.12	12.2	161.38	0.22 ***
	Raffinose	0.67	0.26	9.16	0.01	0.18	3.26	0.99
	Stachyose	2.23	2.55	21.74	0.03	−0.07	2.85	0.99
2020	Sucrose	4.92	4.98	17.2	0.05	−0.13	3.15	0.99
	Raffinose	0.83	0.41	7.28	0.01	0.65	4.83	0.97 ***
	Stachyose	3.61	2.15	9.06	0.02	−0.48	3.8	0.98 **

**Table 2 plants-12-03498-t002:** Two-way ANOVA of seed sugar (sucrose, stachyose, and raffinose) contents in the F × W82 RIL population evaluated in Spring Lake, NC (2018) and Carbondale, IL (2020).

Response: Sucrose				
	**Df**	**Sum Sq**	**Mean Seq**	**H^2^**
Line	369	1134.22	3.0738	0.378
Year	1	5.6	5.5975	
Line × Year	2	3.82	1.9108	
Residuals	0	0	NA	
Response: Raffinose				
	Df	Sum Sq	Mean Seq	H^2^
Line	369	3.4552	0.0093891	0.739
Year	1	0.0253	0.0253139	
Line × Year	2	0.0048	0.0023972	
Residuals	0	0	NA	
Response: Stachyose				
	Df	Sum Sq	Mean Seq	H^2^
Line	369	246.73	0.66865	0.92
Year	1	1.611	1.61115	
Line × Year	2	0.106	0.05307	
Residuals	0	0	NA	

**Table 3 plants-12-03498-t003:** Quantitative trait loci (QTLs) that control sugar (sucrose, stachyose, and raffinose) contents in F × W82 RIL population in Spring Lake, NC in 2018. These QTLs were identified via CIM method. * Indicates novel QTL.

Sugar	QTL	Chr.	Marker/Interval	Position (cM)	LOD	R^2^	Add. Eff.
Sucrose	*qSUC-1*	1	Gm01_3504836-Gm01_3466825	0.01–12.1	39.19	20.46	−3.05
	*qSUC-2*	2	Gm02_5155733-Gm02_9925870	128.5–142.2	42.77	47.90	4.42
	*qSUC-3*	3	Gm03_4595422-Gm03_4113546	39.2–39.8	32.62	20.50	3.05
	*qSUC-4* *	4	Gm04_7672403	6.5–16.5	54.35	37.50	4.62
	*qSUC-5*	5	Gm05_3867435-Gm05_3273418	31.5–37.01	20.65	17.51	2.60
	*qSUC-6*	6	Gm06_1737718-Gm06_5014399	48.5–52.4	5.36	10.50	−1.37
	*qSUC-7*	9	Gm09_1888876	173.9–178.1	32.62	20.50	3.05
	*qSUC-8* *	10	Gm10_621706	214.01–216.01	34.25	19.10	−4.48
	*qSUC-9*	13	Gm13_3891723-Gm13_3524828	0.2–58.2	19.12	17.51	2.60
	*qSUC-10*	17	Gm17_4967175-Gm17_5294475	0.4–1.0	33.22	20.50	3.05
	*qSUC-11* *	18	Gm18_1620585-Gm18_2020823	94.7–96.5	20.10	17.51	2.60
	*qSUC-12*	20	Gm19_2552468	172.11	6.98	9.10	1.41
Stachyose	*qSTA-1*	13	Gm13_3524828	96.2–98.2	2.52	14.8	0.19
	*qSTA-2*	13	Gm13_3884070-Gm13_3803273	121.8–123.2	2.60	5.2	0.11
	*qSTA-3*	19	Gm19_3789399-Gm19_4362616	98.01–124.1	4.21	8.5	−0.16
	*qSTA-4*	19	Gm19_4946208-Gm19_5032228	184.1–186.1	2.53	5.3	0.11
Raffinose	*qRAF-1*	9	Gm09_4024436-Gm09_4082234	108.01–110.9	2.26	4.6	−0.01
	*qRAF-2*	9	Gm09_1888876	173.9–178.1	2.47	7.6	0.08
	*qRAF-3*	12	Gm12_6023395-Gm12_2379195	114.6–118.6	2.15	4.7	−0.01

**Table 4 plants-12-03498-t004:** Quantitative trait loci (QTLs) that control sugar (sucrose, stachyose, and raffinose) contents in F × W82 RIL population in Carbondale, IL in 2020. These QTLs were identified via CIM method.

Sugar	QTL	Chr.	Marker	Position (cM)	LOD	R^2^	Add. Eff.
Sucrose	*qSUC-1*	2	Gm02_1199805-Gm02_1373746	196.4–205.6	2.63	3.60	−0.16
	*qSUC-2*	5	Gm05_3803682-Gm05_3748078	18.01–22.1	2.10	0.03	−0.14
	*qSUC-3*	8	Gm08_5960619-Gm08_8268861	47.1–55.9	2.37	0.04	0.16
Stachyose	*qSTA-1*	13	Gm13_2748576	0.5–4.5	2.03	0.09	0.21
	*qSTA-2*	16	Gm16_3183754-Gm16_3010888	81.6–94.7	2.85	3.92	0.10
	*qSTA-3*	17	Gm17_8449684-Gm17_8352493	136.5–136.7	2.37	3.00	−0.08
	*qSTA-4*	20	Gm20_294157-Gm20_1133712	145.4–148.5	3.59	4.50	−0.12

**Table 5 plants-12-03498-t005:** QTLs and candidate genes that control sugar (sucrose, stachyose, and raffinose) contents in F × W82 RIL population in Spring Lake, NC in 2018. These QTLs were identified via CIM method.

Sugar	QTL	Marker/Interval	LOD	R^2^	Wm82.a2.v1	Start	End	Wm82.a1.v1.1	Start	End	Dis. (MB)
Sucrose	*qSUC-1*	Gm01_3504836-Gm01_3466825	39.19	20.46	.	.	.	.	.	.	.
	*qSUC-2*	Gm02_5155733-Gm02_9925870	42.77	47.9	*Glyma.02G016700*	1490049	1491170	*Glyma02g02030*	1475851	1476528	3.6
	*qSUC-3*	Gm03_4595422-Gm03_4113546	32.62	20.5	.	.	.	.	.	.	.
	*qSUC-4*	Gm04_7672403	54.35	37.5	.	.	.	.	.	.	.
	*qSUC-5*	Gm05_3867435-Gm05_3273418	20.65	17.51	*Glyma.05G040300*	3593378	3598821	*Glyma05g02510*	1870330	1875692	1.3
					*Glyma.05G003900*	307460	312091	*Glyma05g08950*	8806144	8810647	4.9
	*qSUC-6*	Gm06_1737718-Gm06_5014399	5.36	10.5	*Glyma.06G175500*	14845358	14849994	*Glyma06g18480*	14802178	14807061	9.7
					*Glyma.06G179200*	15217419	15223877	*Glyma06g18890*	15175181	15181763	10.16
	*qSUC-7*	Gm09_1888876	32.62	20.5	*Glyma.09G073600*	7809852	7816248	*Glyma09g08550*	7845409	7851685	5.9
					*Glyma.09G016600*	1285132	1290884	*Glyma09g01940*	1270010	1276140	0.6
	*qSUC-8*	Gm10_621706	34.25	19.1	*Glyma.10G017300*	1523661	1524691	*Glyma10g02170*	1519053	1519546	0.8
	*qSUC-9*	Gm13_3891723-Gm13_3524828	19.12	17.51	.	.	.	.	.	.	.
	*qSUC-10*	Gm17_4967175-Gm17_5294475	33.22	20.5	*Glyma.17G037400*	2732048	2737399	*Glyma17g04160*	2739794	2745132	2.2
					*Glyma.17G045800*	3404918	3410491	*Glyma17g05067*	3412682	3418160	1.5
					*Glyma.17G035800*	2629011	2639005	*Glyma17g03990*	2637080	2646732	2.3
					*Glyma.17G111400*	8744555	8747526	*Glyma17g11970*	9015075	9018145	3.7
	*qSUC-11*	Gm18_1620585-Gm18_2020823	20.1	17.51	.	.	.	.	.	.	.
	*qSUC-12*	Gm19_2552468	6.98	9.1	*Glyma.19G004400*	359933	363588	*Glyma19g00441*	238429	242106	2.3
Stachyose	*qSTA-1*	Gm13_3524828	2.52	14.8	.	.	.	.	.	.	.
	*qSTA-2*	Gm13_3884070-Gm13_3803273	2.6	5.2	.	.	.	.	.	.	.
	*qSTA-3*	Gm19_3789399-Gm19_4362616	4.21	8.5	*Glyma.19G004400*	359933	363588	*Glyma19g00440*	241366	241903	3.5
	*qSTA-4*	Gm19_4946208-Gm19_5032228	2.53	5.3	*Glyma.19G004400*	359933	363588	*Glyma19g00440*	241366	241903	4.7
Raffinose	*qRAF-1*	Gm09_4024436-Gm09_4082234	2.26	4.6	*Glyma.09G073600*	7809852	7816248	*Glyma09g08550*	7845409	7851685	3.7
					*Glyma.09G016600*	1285132	1290884	*Glyma09g01940*	1270010	1276140	2.7
					*Glyma.09G167000*	39103764	39109664	*Glyma09g29710*	36530532	36536435	2.5
	*qRAF-2*	Gm09_1888876	2.47	7.6	*Glyma.09G073600*	7809852	7816248	*Glyma09g08550*	7845409	7851685	5.9
					*Glyma.09G016600*	1285132	1290884	*Glyma09g01940*	1270010	1276140	0.6
	*qRAF-3*	Gm12_6023395-Gm12_2379195	2.15	4.7	.	.	.	.	.	.	.

**Table 6 plants-12-03498-t006:** QTLs and candidate genes that control sugar (sucrose, stachyose, and raffinose) contents in F × W82 RIL population in Carbondale, IL in 2020. These QTLs were identified via CIM method.

Sugar	QTL	Marker	LOD	R^2^	Wm82.a2.v1	Start	End	Wm82.a1.v1.1	Start	End	Dis. (MB)
Sucrose	*qSUC-1*	Gm02_1199805-Gm02_1373746	2.63	3.6	*Glyma.02G016700*	1490049	1491170	*Glyma02g02030*	1475851	1476528	0.2
	*qSUC-2*	Gm05_3803682-Gm05_3748078	2.1	0.03	*Glyma.05G040300*	3593378	3598821	*Glyma05g02510*	1870330	1875692	1.8
					*Glyma.05G003900*	307460	312091	*Glyma05g08950*	8806144	8810647	5.002
	*qSUC-3*	Gm08_5960619-Gm08_8268861	2.37	0.04	*Glyma.08G043800*	3450235	3451725	*Glyma08g04860*	3446035	3447462	2.5
					*Glyma.08G143500*	10949673	10956219	*Glyma08g15220*	11038816	11045375	2.7
					*Glyma.08G011800*	942037	944988	*Glyma08g01480*	939512	942346	5.01
					*Glyma.08G023100*	1852651	1856671	*Glyma08g02690*	1848105	1853380	4.1
Stachyose	*qSTA-1*	Gm13_2748576	2.03	0.09	.	.	.	.	.	.	.
	*qSTA-2*	Gm16_3183754-Gm16_3010888	2.85	3.92	.	.	.	.	.	.	.
	*qSTA-3*	Gm17_8449684-Gm17_8352493	2.37	3	*Glyma.17G037400*	2732048	2737399	*Glyma17g04160*	2739794	2745132	5.6
					*Glyma.17G045800*	3404918	3410491	*Glyma17g05067*	3412682	3418160	4.9
					*Glyma.17G035800*	2629011	2639005	*Glyma17g03990*	2637080	2646732	*5.8*
					*Glyma.17G111400*	8744555	8747526	*Glyma17g11970*	9015075	9018145	0.5
	*qSTA-4*	Gm20_294157-Gm20_1133712	3.59	4.5		.	.	.	.	.	.

**Table 7 plants-12-03498-t007:** Candidate genes controlling sugar (sucrose, stachyose, and raffinose) contents associated with previously reported QTLs.

Gene ID	Start	End	QTL	QTL Start	QTL End	Reference
Glyma.02G240400	42892680	42898279	Seed sucrose 2-2	39547350	41441274	[41]
			Seed oligosaccharide 1-1	39547350	41441274	[41]
Glyma.05G236600	41293446	41294570	Seed sucrose 1-1	3924139	4279362	[39]
Glyma.08G043800	3450235	3451725	Seed sucrose 1-3	7892162	8937354	[39]
Glyma.08G143500	10949673	10956219	Seed sucrose 1-2	10865328	13126779	[39]
Glyma.09G073600	7809852	7816248	Seed sucrose 4-2	2973041	5901485	[44]
Glyma.13G114000	22767704	22773231	Seed sucrose 1-5	26196486	28912864	[39]
Glyma.14G209900	47515899	47521687	Seed sucrose 3-1	38859467	40060720	[40]
			Seed oligosaccharide 2-1	38859467	40060720	[40]
Glyma.15G151000	12497113	12508050	Seed sucrose 3-3	13755345	17021739	[40]
			Seed oligosaccharide 2-3	13755345	17021739	[40]
Glyma.19G140700	40199041	40201038	Seed sucrose 1-8	40205349	40265091	[39]
			Seed oligosaccharide 2-7	42119600	43329204	[40]
Glyma.19G212800	46633685	46639818	Seed oligosaccharide 2-7	42119600	43329204	[40]
			qSU1901	45311975	45464136	[43]
Glyma.19G217700	47033812	47037286	Seed oligosaccharide 2-7	42119600	43329204	[40]
			qSU1901	45311975	45464136	[43]
Glyma.20G095200	33827363	33831352	Seed sucrose 1-4	2716974	25498552	[39]
Glyma.08G011800	942037	944988	Seed sucrose 1-3	7892162	8937354	[39]
			Seed sucrose 1-13	8283676	9192408	[39]
Glyma.08G023100	1852651	1856671	Seed sucrose 1-3	7892162	8937354	[39]
			Seed sucrose 1-13	8283676	9192408	[39]
Glyma.19G219100	47148224	47150373	Seed sucrose 1-8	40205349	40265091	[39]
			Seed sucrose 2-10	40637071	41616190	[41]
			Seed sucrose 2-11	40637071	41616190	[41]
			Seed oligosaccharide 2-7	42119600	43329204	[40]
Glyma.19G227800	47911129	47914214	Seed sucrose 1-8	40205349	40265091	[39]
			Seed sucrose 2-10	40637071	41616190	[41]
			Seed sucrose 2-11	40637071	41616190	[41]
			Seed oligosaccharide 2-7	42119600	43329204	[40]
Glyma.20G094500	33759416	33761555	Seed sucrose 1-4	2716974	25498552	[39]
Glyma.20G177200	41446962	41451980	qSU2002	40523599	41882459	[43]
Glyma.15G182600	17910130	17916426	Seed sucrose 3-3	13755345	17021739	[40]
			Seed oligosaccharide 2-3	13755345	17021739	[40]
Glyma.05G003900	307460	312091	Seed sucrose 1-1	3924139	4279362	[39]
Glyma.09G016600	1285132	1290884	Seed sucrose 4-2	2973041	5901485	[44]
Glyma.17G111400	8744555	8747526	qSS1701	7470395	10014816	[43]
			qSS1702	7969537	10599548	[43]
Glyma.13G160100	27576191	27579282	Seed sucrose 1-5	26196486	28912864	[39]
Glyma.19G004400	359933	363588	Seed sucrose 2-3	4244065	12744826	[41]
			Seed oligosaccharide 1-2	4244065	12744826	[41]
			Seed sucrose 2-6	9284015	34059981	[41]
			Seed oligosaccharide 1-5	9284015	34059981	[41]

## Data Availability

The data presented in this study are available upon request from the corresponding author.

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
