# Peer review of "Quantitative Trait Loci and Candidate Genes That Control Seed Sugars Contents in the Soybean ‘Forrest’ by ‘Williams 82’ Recombinant Inbred Line Population"

_plants, 2023, doi:10.3390/plants12193498_

Round 1

Reviewer 1 Report

In this study, the authors conducted QTL mapping on a RIL population to dissect the genetic basis of seed sugar contents in soybean, employing both classical forward genetic techniques and reverse genetic approaches to identify candidate genes. However, there are several concerns that should be addressed before the study can be considered for acceptance:

(1) One of the primary concerns in this study pertains to the selection of candidate genes based on a physical distance of 10/20MB surrounding the identified QTL. It is worth noting that this physical distance is not a reasonable choice given the recombination rate in a soybean RIL population. To enhance the robustness of the findings, it is essential to consider a more suitable genetic distance, instead of the huge physical distance. The inclusion of a physical region greater than 40MB, exceeding the size of most soybean chromosome arms, raises questions about its relevance to the QTL results. Alternatively, presenting the QTL mapping and sugar pathway gene analyses as distinct sections, even if there is limited overlap between them, may be a viable solution.

(2) Heritability is a critical factor influencing QTL mapping outcomes. It would be beneficial for the authors to discuss the observed variations in broad-sense heritability across the three sugar traits, as presented in Table 2. Of particular interest is the detection of more QTL with higher LOD values in traits with lower heritability, as shown in Table 3. Addressing this phenomenon in the discussion would provide a more comprehensive understanding of the results.

(3) To enhance the clarity of the QTL mapping process, the authors should provide information about the polymorphic molecular markers used for anchoring the genetic map. Additionally, including the genetic map as a primary figure, illustrating the markers and identified QTL, would greatly aid in visualizing and interpreting the research findings.

(4) In the context of seed sugar quantification, the study should provide more detailed information regarding the number of replicates used for each RIL and the parental lines. This transparency is vital for evaluating the reliability and reproducibility of the results.

(5) Figure 1: It is advisable to enhance the readability of Figure 1 by increasing the font size of the axis labels, ensuring that the graphical presentation is more accessible to the reader.

(6) Maintaining consistent formatting throughout the manuscript is essential. The use of italics in the main text should also be extended to Table 7 and Figures 3 and 4 to maintain uniformity and improve overall presentation.

Reviewer 2 Report

The objective of this study was to genetically map QTL for seed sucrose, raffinose, and stachyose contents using the ‘Forrest’ by ‘Williams 82’ RIL population, in addition to identifying candidate genes involved in soybean seed sugars biosynthesis. Although the authors attempted to identify candidate genes by reverse BLAST, expression analysis and sequence alignment. However, due to the low resolution of the QTL mapping, this study did not get enough credible and valuable information for breeding programs of reducing raffinose and stachyose and increasing sucrose in soybean seed content.

Line 74: “Only one of these studies identified candidate genes within these QTL regions” but two references are listed.

Lines 141-142: How to explain that “The histogram of sucrose 2018 was extremely skewed, and the other traits evaluated were normally distributed” ?  Accurate phenotype data is crucial for QTL mapping. How did the author ensure the accuracy of the mapping results by using abnormal phenotype data?

In the section of discussion, the author re-describes his research results in large paragraphs (lines 135-181), lacking conclusive or extensibility arguments. 

The author seems to have written this paper in a hurry. For example, the format of the references is somewhat confusing, and the font sizes are inconsistent in lines 90-92. The line numbers of the manuscript were also repeatedly numbered several times. The mistakes listed above gave me some uncomfortable reviewing experiences. It is suggested that the author seriously revise the language and words of this manuscript.

Round 2

Reviewer 1 Report

All concerns are addressed. I have no more comments. 

Author Response

Thank you very much!

Reviewer 2 Report

Figure 1:  It is recommended to increase the number of groups in the frequency distribution of sucrose in 2018, which may be possible to show the phenotype distribution more clearly.

I believe that the phenotypic data of sucrose in 2018 you present in your manuscript are accurate. However, according to the histogram of phenotype distribution, it can be seen that the sugar content of the two parents, especially the sucrose content in 2018, is not very different, which means that there is probably no differential allele controlling the target trait between the two parents. It is recommended that the authors provide an appropriate explanation of the phenotypic results and the feasibility of using mapping populations derived from parents with small differences in the target trait for QTL mapping in the later discussion section.

Author Response

See comments on the attached response.